

# Sensitive detection of *n*-alkanes using a mixed ionization mode Proton-Transfer Reaction – Mass Spectrometer

Omar Amador-Muñoz[1,2], Pawel K. Misztal[1], Robin Weber[1], Dave R. Worton[3], Haofei Zhang[1], Greg Drozd[1], Allen H. Goldstein[1]

[1] Department of Environmental Science, Policy, and Management, University of California, Berkeley, CA, USA.
[2] Centro de Ciencias de la Atmosfera, Universidad Nacional Autonoma de Mexico, Mexico.
[3] National Physical Laboratory, Teddington, Middlesex, TW11 0LW, U.K.

*Correspondence to*: Omar Amador-Muñoz (oam@atmosfera.unam.mx)

**Abstract.** Proton transfer reaction mass spectrometry (PTR-MS) is a technique that is widely used to detect volatile organic compounds (VOCs) with proton affinities higher than water. However, *n*-alkanes generally have a lower proton affinity than water and therefore proton transfer (PT) by reaction with $H_3O^+$ is not an effective mechanism for their detection. In this study, we developed a method using a conventional PTR-MS to detect *n*-alkanes by optimizing ion source and drift tube conditions to vary the relative amounts of different primary ions ($H_3O^+$, $O_2^+$, $NO^+$) in the reaction chamber (drift tube). We determined the optimum conditions and the resulting reaction mechanisms allowing detection of *n*-alkanes from *n*-pentane to *n*-tridecane. These compounds are mostly emitted by evaporative/combustion process from fossil fuel use. The charge transfer (CT) mechanism observed with $O_2^+$, was the main reaction channel for *n*-heptane and longer *n*-alkanes, while for *n*-pentane and *n*-hexane, the main reaction channel was hydride abstraction (HA). Maximum sensitivities were obtained at low E/N ratios (83 Td), low water flow (2 sccm) and high $O_2^+/NO^+$ ratios ($U_{so}$ = 180 V). Isotopic $^{13}C$ contribution was taken into account by subtracting fractions of the preceding $^{12}C$ ion signal based on number of carbon atoms and the natural abundance of $^{13}C$ (i.e. 5.6 % for n-pentane and 14.5 % for n-tridecane). After accounting for isotopic distributions, we found that proton transfer cannot be observed for n-alkanes smaller than *n*-decane. Instead, protonated water clusters of *n*-alkanes ($M·H_3O^+$) species were observed with higher abundance at lower $O_2^+$ and higher water clusters amount fractions. $M·H_3O^+$ species are probably the source for $M+H^+$ species observed from *n*-decane to *n*-tridecane, via a collision dissociation mechanism. Normalized sensitivities to $O_2^+$ or to the sum of $O_2^++NO^+$ were determined to be a good metric with which to compare sensitivities for *n*-alkane detection between experiments. Double hydride abstraction was observed from the reaction with $O_2^+$. Sensitivities to CT and HA both increased with carbon chain length from *n*-pentane to *n*-dodecane while PT sensitivity increased from *n*-decane to *n*-tridecane. Sensitivity to charge transfer exponentially decreased with molecular ionization energy which is inversely related to the carbon chain length. We define optimum conditions for using mixed ionization mode to measure *n*-alkanes in conventional PTR-MS instruments that are not equipped with switchable reagent ions (SRI) capabilities.

**Keywords:** Charge transfer, Hydride abstraction, n-alkanes, $O_2^+$, $NO^+$



## 1  Introduction

Proton transfer reaction mass spectrometry (PTR-MS) has been extensively used to measure volatile organic compounds (VOCs) (e.g. de Gouw and Warneke, 2007). PTR-MS is normally optimized for just $H_3O^+$ ionization but that does not allow efficient detection of *n*-alkanes and some other species that are extremely important for air quality, atmospheric chemistry, and source emission characterization. This technique offers close to real time data acquisition and low detection limits (low pptv) (Hansel et al., 1995). Chemical ionization (CI) from different reagent ions is also now possible in PTR-MS instruments equipped with Switchable Reagent Ions (SRI) and with Selected Ion Flow Tube Mass Spectrometers (SIFT-MS). CI is achieved mainly through reaction with an atomic or molecular ion following four possible reaction processes: charge transfer, proton transfer, hydride abstraction or adduct formation (Ellis and Mayhew, 2014). The reactions of $H_3O^+$ with VOCs (R.1) are thermodynamically favored when the proton affinity of its conjugate base, $H_2O$, is lower ($691\pm3$ kJ mol$^{-1}$) than that of the VOCs in question (Hunter and Lias, 1998). However, the proton affinities for commonly encountered inorganic compounds in air, such as nitrogen and oxygen, and also for some haloalkanes and alkanes, are higher and therefore they are not detectable by proton transfer (R.1).

$$H_3O^+ + C_nH_{2n+2} \mid n\geq10 \longrightarrow C_nH_{2n+2}+H^+ + H_2O \qquad \text{Proton Transfer (PT)} \qquad \text{R.1}$$

Alkanes are among the most abundant compounds in urban air (Amador-Muñoz et al., 2011). Petroleum derived air pollution sources, including gasoline and diesel fuels (Gentner et al., 2012; 2013), motor oils (Worton et al., 2014) or crude oil (Drozd et al., 2015; Worton et al., 2015), contain significant amounts of alkanes. These compounds react in the atmosphere to produce ozone and secondary organic aerosol (Jordan et al., 2008; Lim and Ziemann, 2005) with implications for climate and human health. Adapting PTR-MS to measure alkanes simultaneously along with other VOCs is therefore of significant interest. Previous studies have been conducted to explore measurement of *n*-alkanes in PTR-MS experiments (Erickson et al., 2014; Jobson et al., 2005) in conventional $H_3O^+$ mode. However, proton transfer reaction products are mostly unobserved for short-chain alkanes because the reaction does not proceed at the collisional rate, although the proton-transfer reaction of $H_3O^+$ with *n*-alkanes larger than *n*-hexane is expected to be exothermic (Arnold et al., 1998). Instead, association is the predominant reaction mechanism observed at room temperature for *n*-alkanes (Arnold et al., 1998; Španěl and Smith, 1998).

*N*-alkanes ($N_c$=2-12) can react with $NO^+$ typically through hydride abstraction (Arnold et al., 1997; 1998; Inomata et al., 2013; Lias et al., 1976; Searles and Sieck, 1970, Španěl and Smith, 1998; Wilson et al., 2003;) (R.2)

$$NO^+ + C_nH_{2n+2} \longrightarrow C_nH_{2n+1}^+ + HNO \qquad \text{Hydride abstraction (HA)} \qquad \text{R.2}$$

While reactions with $O_2^+$ have been observed to proceed by both charge transfer (R.3) ($N_c$=2-12) (Arnold et al., 1997; Francis et al., 2007; Španěl and Smith, 1998; Wilson et al., 2003), hydride abstraction (R.4) ($N_c$=2-4) (Francis et al., 2007; Wilson et al., 2003) and double hydride abstraction (R.5) ($N_c$=2-3) (Francis et al., 2007; Wilson et al., 2003).

$$O_2^+ + C_nH_{2n+2} \longrightarrow C_nH_{2n+2}^+ + O_2 \qquad \text{Charge transfer (CT)} \qquad \text{R.3}$$

$$O_2^+ + C_nH_{2n+2} \longrightarrow C_nH_{2n+1}^+ + (O_2 + H) \qquad \text{Hydride abstraction (HA)} \qquad \text{R.4}$$





$O_2^+ + C_nH_{2n+2} \longrightarrow C_nH_{2n}^+ + (O_2 + 2H)$      Double Hydride abstraction (DHA)      R.5

However, reactions with $O_2$ are often dissociative because the ionization energy of $O_2$ (12.07 eV; NIST Chemistry Web Book), is higher than the vast majority of organic molecules, leading to significant fragmentation (Blake et al., 2006; Wyche et al., 2005). For this reason, application of $O_2^+$ ionization has so far been limited in the PTR-MS community. It is possible to suppress fragmentation by keeping the field strength (E/N, where E is the electric field strength and N the buffer gas density) in the drift tube sufficiently low. Typical PTR-MS conditions have $O_2^+$
impurities typically ranging from 1-5 % of $H_3O^+$ (de Gouw and Warneke, 2007). Recently, the PTR-MS instrument has been equipped with an SRI capability, which allows for switching between more pure $H_3O^+$, $NO^+$, and $O_2^+$ ion modes (Jordan et al., 2009; Karl et al., 2012; Inomata et al., 2013).

In this study we show how a PTR-MS (without SRI capability) can successfully be optimized to measure *n*-alkanes
and other major VOCs at the same time, by making $O_2^+$, $H_3O^+$ and $NO^+$ reagent ions simultaneously available. We also define the reaction mechanisms leading to detection of *n*-alkanes using these three reactant ions.

## 2 Methods

### 2.1 Proton transfer reaction - mass spectrometer

PTR-MS is a highly sensitive technique for quantification of atmospheric VOCs close to real time (10 Hz) (Karl et al., 2002). The operation and fundamental principles of PTR-MS have been described in detail previously (Hansel et al., 1995; Lindinger et al., 1998), and therefore only a brief description is given here. The three main components of the PTR-MS are the ion source, drift tube and detector. In conventional PTR-MS, the ion source is a hollow cathode that produces hydronium ion ($H_3O^+$), using a plasma discharge and water ($H_2O$). During this process, some $O_2^+$ and
$NO^+$ ions are also produced. The extracted ions from the ion source are accelerated in the drift tube by an electrical field (E). The air containing VOCs to be analyzed operates as buffer gas (N) and is also introduced into the drift tube. Reagent ions and VOCs selectively collide under specific E/N ratios, maintaining a buffer gas pressure of typically 2-2.3 mbar. The proton transfer and any other reactions occur in the drift tube and their product ions are filtered by a quadrupole mass spectrometer and counted by a secondary electron multiplier (SEM).
Different reactions in the drift tube can be favored according to the abundance and ratios among the ions generated in the ion source. The PTR-MS ion source is typically optimized so that $H_3O^+$ is the most abundant ion. However, water can be substituted by pure $O_2$ or NO to produce $O_2^+$ or $NO^+$ as the dominant ions, respectively, as is the case in instruments equipped with SRI (Inomata et al., 2013; Jordan et al., 2009; Karl et al., 2012). In a conventional PTR-MS (without SRI) the relative abundance of $O_2^+$ and $NO^+$ ions can be enhanced compared to $H_3O^+$ by
increasing the source out voltage ($U_{SO}$).

We optimized ion source and drift tube conditions to vary the relative amounts of ions produced using a conventional PTR-MS to determine the best conditions for detecting *n*-alkanes with different carbon chain lengths. Drift-tube temperature and drift-tube pressure were kept constant at 60 °C and at 2.00 ±0.02 mbar, respectively. Mass scans were recorded from 20 to 220 Da (200 ms dwell time) for 45 cycles in each experiment. This range
included the $H_3O^+$ isotopic primary ion (m/z 21), $NO^+$ ion (m/z 30), isotopic $O_2^+$ ion (m/z 34) and water clusters (mono-hydrated, m/z 37 and bi-hydrated, m/z 55). $O_2^+$ count rates were calculated from the isotopic oxygen signal at



m/z 34, times 250; $H_3O^+$ counts were calculated from the isotopic contribution at m/z 21, times 500. Optimization tests included adjusting the amount of water supplied to the ion source, adjusting the drift tube voltage, and adjusting the source out voltage. Water flow to the ion source to generate $H_3O^+$ was tested at two conditions: low

water flow (LWF) at 2 sccm and high water flow (HWF) at 6 sccm. The field strength of the drift tube, *E/N*, was tested under 5 sets of conditions including 83, 91, 101, 109 and 122 Td (1 Td = $10^{-17}$ V cm$^2$ molecule$^{-1}$), which were achieved by setting the drift tube ($U_{drift}$) voltage to 343, 375, 419, 461 and 505 V, respectively. To vary $O_2^+$ reagent ion in the hollow cathode discharge ion source, the source out voltage ($U_{SO}$) was cycled between 60, 90, 120, 150 and 180 V.


### 2.2 Experimental setup and methods

All measurements reported here were done using a PTR-MS with a quadrupole mass spectrometer (Ionicon AnalytiK). Pure standards of nine *n*-alkanes ranging from *n*-pentane to *n*-tridecane were used. *N-p*entane (≥ 99%), *n-h*exane (≥ 99%), *n-d*ecane (≥ 99%) and *n-u*ndecane (≥ 99%) were purchased from Sigma-Aldrich. *N-h*eptane (99 %)

from Spectrum. *N-o*ctane (>99 %) and *n*-tridecane (> 99%) from Acros Organics and *n*-nonane (99%) and *n*-dodecane (>99 %) from Alfa Aesar. Water used for producing $H_3O^+$ ions was obtained from BDH Reagents & Chemicals. Table 1 shows the molecular formula, molecular weight, $^{13}$C isotopic abundance and ionization energies for these nine *n*-alkanes along with the monitored ions for each reaction mechanism. These *n*-alkane standards were introduced into a custom built flow tube system (Zhang et al., 2013) to mix and dilute the *n*-alkanes with zero air

(Fig. 1). The standards in liquid form were injected with a syringe pump (Kd Scientific) at 0.2 μL min$^{-1}$ into a zero air stream produced by a zero air generator (Aadco 737) and introduced into the flow tube throughout a silcosteel transfer line heated to 50 ± 4 °C. A total flow of 2.3 L min$^{-1}$ was maintained through the flow tube for the experiments. Two set of experiments were conducted: a first set of experiments to determine reaction mechanisms and optimize conditions for reaction with different ions, and a second set of experiments to determine sensitivities

for individual *n*-alkanes under optimized conditions.

### 2.2.1 Optimization experiments

To determine the reaction mechanisms of *n*-alkanes in the PTR-MS and to optimize conditions for their detection, a constant flow of 40 mL min$^{-1}$ (MKS flow controller) of *n*-hexane (1340 ppbv), *n*-decane (445 ppbv), n-nonane (482

ppbv), *n*-undecane (406 ppbv) and *n*-dodecane (377 ppbv) were introduced into the flow tube. The mechanism of reaction was determined from the observed m/z ratios. Proton transfer was indicated by the addition of 1 amu to the parent mass (M+H), charge transfer by detection of the parent mass (M), hydride abstraction by the loss of 1 amu (M-H), and double hydride extraction by the loss of 2 amu (M-2H). Relative humidity (RH) was controlled at < 2 %. RH and room temperature were recorded every 30 seconds (Humitter 50Y, Vaisala). Sensitivities (cps/ppbv) to

charge transfer, hydride abstraction and proton transfer reactions were determined for all 50 different combinations of E/N ratios, primary ions abundances, and water flow conditions. Sensitivities were corrected by subtracting the isotopic $^{13}$C contribution of the preceding $^{12}$C ion (Table 1).



### 2.2.2 Sensitivity determinations

To determine PTR-MS measurement sensitivities using optimized conditions under typical ambient humidity conditions, calibration curves were generated by introducing each individual $n$-alkane into the flow tube at three different flow rates (20, 50 and 80 mL min$^{-1}$) while the total flow was maintained constant at 2.3 L min$^{-1}$. These experiments were conducted at room temperature with a relative humidity of 30 %. Sensitivities were corrected by

subtracting the isotopic $^{13}C$ contribution of the preceding $^{12}C$ ion (Table 1). Table S1 shows the concentration ranges generated for each $n$-alkane.

## 3   Results

### 3.1 Relative ion abundances used to determine $n$-alkane reaction mechanisms

The relative abundance of reactant ions varied as a function of E/N ratios, $U_{SO}$ voltages and water flows. Under normal PTR-MS conditions optimized for $H_3O^+$, there are still $O_2^+$ and $NO^+$ ions produced as byproducts in the ion-source discharge. In our experiments, conditions were adjusted to enhance the relative amounts of $O_2^+$ and $NO^+$ by decreasing the water flow from 6 to 2 sccm and increasing the $U_{SO}$ from 60 to 180 V (Fig. S1). The optimal flow-rate of 2 sccm was chosen based on the relatively highest $O_2^+/H_3O^+$ ratio as well as the total absolute $O_2^+$ ion

abundance at non-saturating conditions. Depending on the type of the ion source in other instruments and the ion source cleanliness, the actual $H_2O$ flow rates may result in different equilibria. $O_2^+$ percentage relative to $H_3O^+$ in LWF ranged from 34 to 270 % ($U_{SO}$ from 60 to 180 V), and in HWF from 5 to 68 %. $NO^+$ relative to $O_2^+$ observed in LWF ranged from 6 and 21 %, and in HWF from 1 to 4 %. $O_2^+$ count rates were on average around 10 times higher than $NO^+$ count rates. With HWF and low E/N, excessive water clusters were observed and the primary ion

($H_3O^+$) was saturated.

### 3.2 Maximizing sensitivity and determining dominant reaction mechanisms

     The dominant reaction mechanism occurring for different $n$-alkanes, and also the overall sensitivities of detection, varied as a function of the relative abundance of different primary ions, and could be controlled by varying $U_{SO}$,

water flow, and E/N ratio. The absolute sensitivities varied substantially with $U_{so}$ for different $n$-alkanes, while low E/N ensured the highest sensitivity for all the $n$-alkanes. Maximum sensitivities from $n$-nonane to $n$-dodecane were obtained at LWF, the lowest E/N ratios (83 Td), and the maximum $U_{SO}$ (180 V). Figure 2 shows the sensitivities (cps/ppbv) for two example $n$-alkanes: $n$-hexane and $n$-decane. Three ions were plotted for $n$-hexane and for $n$-decane, produced from the HA, CT and PT mechanisms. The highest sensitivity for n-hexane was obtained by HA

(m/z 85) followed by CT (m/z 86). PT (m/z 87) sensitivities were negligible (< 1 % with respect to HA). In the case of n-decane, maximum sensitivities were obtained by CT (m/z 142), followed by HA (m/z 141). Sensitivities by PT were < 8 % with respect to CT. $N$-nonane, $n$-undecane and $n$-dodecane showed the same sensitivity trend as $n$-decane (Fig. 2b).



Table 2 shows the comparisons in sensitivities for the different mechanisms. We also compared these sensitivities
under "typical" PTR-MS conditions to analyze VOCs: water flow 6 sccm, E/N ratio 122 Td and $U_{SO}$ 60 V.
Sensitivities for some ions in "typical" conditions were close to zero. However, under optimized conditions,
sensitivities of *n*-hexane by HA were 344 times higher than sensitivities by PT, and 1.9 times higher than by CT. In
the case of higher *n*-alkanes, sensitivities by CT were higher than by PT but the ratios decreased consistently with
the chain length, with higher PT for larger *n*-alkanes due to increasing proton affinities. A constant effect throughout
the chain length was observed for the sensitivities ratios between CT and HA. Table 2 shows why under typical
PTR-MS conditions, *n*-alkanes are not detected by PT. Maximum sensitivities were observed by CT and HA when
water flow was significantly reduced (2 sccm in this study), thus further discussion will deal only with this low flow
condition and will focus on $O_2^+$ and $NO^+$ ionizations.

### 3.2.1 $O_2^+$ and $NO^+$ ionizations

Molecules with ionization energies lower than the recombination energy of $NO^+$ (9.26 eV) may undergo charge
transfer (Ellis and Mayhew, 2014; Lias, 2000), for example, aromatics and alkenes (Karl et al., 2012; Liu et al.,
2013). *N*-alkanes in this study have ionization energies (Table 1) higher than $NO^+$, so no charge transfer was
expected. Therefore, charge transfer theoretically should be exclusively due to $O_2^+$. Figure 3a illustrates strong
positive associations between $O_2^+$ and the signal of n-decane in CT (m/z 142, r≥ 0.98). Similar behavior was
observed for *n*-nonane, *n*-undecane and *n*-dodecane (Fig. S2). Reactions between $O_2^+$ and *n*-alkanes have been
suggested to result from the CT mechanism (Arnold et al., 1997; Francis et al., 2007; Španěl and Smith, 1998;
Wilson et al., 2003) at or close to the collisional rate (Lias et al., 1976; Searles and Sieck, 1970), except for methane
(Barlow et al., 1986).

The relative abundance of the molecular ion reaction product increased with $O_2^+$, and decreased with E/N, due to
higher fragmentation (Fig. S3). Fragments m/z 57 $[C_4H_9]^+$, m/z 71 $[C_5H_{11}]^+$, m/z 85 $[C_6H_{13}]^+$, m/z 99 $[C_7H_{15}]^+$ and
m/z 113 $[C_8H_{17}]^+$ were the most abundant at lower E/N ratios, but m/z 43 $[C_3H_7]^+$ and in particular m/z 29 $[C_2H_5]^+$
were the most abundant at higher E/N. This was observed for all n-alkanes. CT in $O_2^+$ is dissociative and produces
multiple product ions (Španěl and Smith, 1998). The identification of n-alkanes or other compounds in highly
complex mixtures using the actual conditions is difficult, and therefore has not yet been widely utilized. However,
we comprehensively approach these challenges and show that varying E/N ratio results in different proportions of
these common fragments. The interferences and sensitivities are always a general issue for all ionization modes and
in particular for quadrupole detectors and even more when fragmentation is high. While this can be an issue at trace
levels of n-alkanes in biogenically influenced air (where isoprene oxidation products can result in the same nominal
masses), this should not be a problem for characterizing crude oil evaporation or alkane presence in pollution
plumes dominated by n-alkanes. Potential interferences could be from alkenes but they are not present in the crude
oil and their presence and abundance can also be inferred from different mechanisms.

$O_2^+$ has been reported to produce significant yields of HA species for ethane (55 %) and propane (40 %) (Francis et
al., 2007), although Arnold et al. (1997) reported 2 % $[C_8H_7^+]$ for *n*-octane. Unlike in the case of $O_2^+$, $NO^+$ has been



shown to react by HA for $\geq C_5$ *n*-alkanes (Arnold et al., 1997; 1998; Španěl and Smith, 1998). This is illustrated in Fig. 3b were high correlations were obtained between products of HA and $NO^+$ (r=0.97-0.99, E/N $\leq$109 Td). Because in our study $NO^+$ was mixed with $O_2^+$ (1:10) we cannot distinguish the dominant ion for HA using PTR-MS.

Enthalpy is one reason to explain the low sensitivities of $[M-H]^+$ species for smaller *n*-alkanes with $NO^+$. Formation of $[M-H]^+$ by reaction between $NO^+$ and *n*-alkanes ($\leq C_5$) has been reported to be endothermic (Hunt and Harvey, 1975; Arnold et al., 1998). In addition, the reaction rate constants decrease with carbon chain length for the *n*-alkanes (Arnold et al., 1998; Hunt and Harvey, 1975; Lias et al., 1976; Searles and Sieck, 1970; Španěl and Smith, 1998, Wilson et al., 2003). Arnold et al. (1998) observed 20 % yields for $[M-H]^+$ species in the reaction of n-hexane with $H_3O^+$, while < 5 % was observed in the reactions with *n*-heptane to *n*-decane. Similarly, Wilson et al. (2003) reported yields of 30 % of $[M-H]^+$ species on *n*-butane with $H_3O^+$. However, the rate constant reported by Arnold et al. (1998) was 0.006 x $10^{-9}$ $cm^3$ $s^{-1}$, while Wilson et al. (2003) did not report the rate constant. Španěl and Smith (1998) and Francis et al. (2007) did not report $[M-H]^+$ species from reaction between n-alkanes and $H_3O^+$. In our study we did not find any correlation between $H_3O^+$ signal with hydride abstraction species for any of the *n*-alkanes studied.

### 3.2.2 $H_3O^+$ effects

In our study, $[M+H]^+$ species were not observed for *n*-alkanes smaller than *n*-decane. For larger *n*-alkanes, a small signal intensity for $[M+H]^+$ was observed, which increased with increasing carbon chain length (Figs. 2, S1). The formation of $[M+H]^+$ species for short-chain *n*-alkanes is difficult because the reaction does not proceed at the collisional rate, but the proton-transfer reaction of $H_3O^+$ with *n*-alkanes larger than *n*-hexane is expected to be exothermic (Arnold et al., 1998).

The $[M+H]^+$ did not correlate with $H_3O^+$, indicating it is not the proton donor, suggesting no significant direct proton transfer occurs between $H_3O^+$ and *n*-alkanes in this system. The same results were reported by Arnold et al. (1998), who observed no direct proton transfer product ions in the form $C_nH_{2n+3}^+$ (*Nc*=2-12) in their SIFT experiments. Španěl and Smith (1998) also did not observe protonated parent alkanes (*Nc*=4-12), apparently due to their instability in SIFT experiments. No direct proton transfer reaction is expected for formation of $[M+H]^+$ from *n*-alkanes by PT from water clusters (Jobson et al., 2005), because the proton affinities for these compounds are lower than the water cluster.

Instead of direct proton transfer to *n*-alkanes, $[M \cdot H_3O]^+$ species have been observed to be produced through an association mechanism between *n*-alkanes and $H_3O^+$ primary ion (Jobson et al., 2005; Španěl and Smith, 1998) (R.6). Španěl and Smith (1998) observed in SIFT experiments that reaction of $H_3O^+$ with *n*-alkanes, occurs near the collisional limit to form only association product ions as $[M \cdot H_3O]^+$. These species occur when the proton affinity is less than the proton affinity of water, and proton transfer is endothermic (Smith and Španěl, 2005). We observed $[M \cdot H_3O]^+$ species formed from n-alkanes as illustrated by the adduct of *n*-decane at m/z 161 (Fig. S3). Positive correlations between hydronium-decane adduct $[C_{10}H_{22}.H_3O]^+$ (m/z 161) with $H_3O^+$ (Fig. 4a) were observed. Similar





correlations were found with $H_2O.(H_3O)^+$ (R.7). Higher $H_3O^+$ is produced at lower $O_2^+$, and therefore, more $[M \cdot H_3O]^+$ was observed under these conditions. Similarly, a higher abundance of water clusters are produced at lower E/N ratio, and more $[M \cdot H_3O]^+$ was observed at lower $O_2^+$. Once the $[M \cdot H_3O]^+$ is formed, a collisional dissociation can take place producing $[M+H]^+$ (R.8., Fig. 4b). An inverse relationship between $[C_{10}H_{22}.H_3O]^+$ and $[M+H]^+$ (m/z 143) was observed. At higher $O_2^+$, more de-clustering occurs decreasing $[M \cdot H_3O]^+$ and increasing $[M+H]^+$.

Therefore, we propose the $[M+H]^+$ comes from the adduct formation from $[M \cdot H_3O]^+$ found for these compounds (R.6/R.7) and a subsequent collisional dissociation (R.8).

| | | | | |
|---|---|---|---|---|
| $H_3O^+ + C_{10}H_{22}$ | $\longrightarrow$ | $[C_{10}H_{22}.H_3O]^+$ | Adduct formation | R.6 |
| $H_2O.(H_3O)^+ + C_{10}H_{22}$ | $\longrightarrow$ | $[C_{10}H_{22}.H_3O]^+ + H_2O$ | Adduct formation | R.7 |
| $[C_{10}H_{22}.H_3O]^+ + M$ | $\longrightarrow$ | $[C_{10}H_{22}.H]^+ + H_2O + M$ | Collisional dissociation | R.8 |
| $[C_{10}H_{22}.H_3O]^+ + H_2O$ | $\longrightarrow$ | $H_2O.(H_3O)^+ + C_{10}H_{22}$ | Ligand switching | R.9 |

Španěl and Smith (1998) observed ligand switching between $[M.H_3O]^+$ and the $H_2O$ in SIFT experiments when a trace of water vapor was present in the carrier gas (R.9). This can catalyze the production of the hydrated hydronium ion via R.6 and R.9. Switching reactions usually occur at the collisional rate, including reactions with alkanes (Španěl and Smith, 1998). Maximum $[M.H_3O]^+/M^+$ percentages were observed in higher E/N ratios and lower USO. The stability of these species is probably due to an ion-induced dipole interaction strengthened by the higher polarizability of larger alkanes (Anslyn and Dougherty, 2006); increasing the bond strength of the cluster (Arnold et al., 1998). The rates of such association reactions apparently increase as the proton affinity approaches that of water (Španěl and Smith, 1998).

### 3.3 Sensitivities and Linearity of response

The optimal PTR-MS conditions were applied to determine the response linearity of the *n*-alkanes. Experiments were carried out at 83 Td, $U_{SO}$ = 180 V and $H_3O^+$ = 1 sccm. We extended the number of *n*-alkanes from *n*-pentane to *n*-tridecane. Table S1 indicates the range of concentrations tested.

Since the sensitivity (cps/ppbv) of target compounds is a function of the primary ion abundance, and the absolute values of water flow in different PTR-MS instruments may result in different levels of $O_2^+$, normalized sensitivity (ncps/ppbv) is recommended to account for their variability (Jobson et al., 2005; Warneke et al., 2001). $H_3O^+$ is normally used when proton transfer is the dominant mechanism. However, in our study, we calculated four types of sensitivities: absolute, normalized to $O_2^+$, to $NO^+$ and to the sum of $O_2^+ + NO^+$, for all *n*-alkanes. To determine which approach is the best metric for sensitivity comparisons, we compared the sensitivities for *n*-nonane, *n*-decane, *n*-undecane and *n*-dodecane obtained in the first and the second set of experiments. These *n*-alkanes followed exactly the same mechanisms among them, as described earlier. Figure S4 shows stabilities of sensitivities. Normalized sensitivities to $O_2^+$ resulted in the minimum variability in sensitivities by HA reactions, while the normalization to the sum of $[O_2^+ + NO^+]$ resulted in the minimum variability in sensitivities by CT reactions. Results suggest that normalization to both ions makes sense for CT and that normalization to $O_2^+$ makes the most sense for HA.



Figure 5 shows the normalized sensitivities for all nine *n*-alkanes determined in the optimized conditions. As we observed previously in the first set of experiments, CT was the dominant mechanism for *n*-heptane and larger *n*-alkanes, while HA was dominant for *n*-hexane and smaller *n*-alkanes. The limits of detection (LODs) are dependent on the sensitivity and standard deviation of the VOC-free background (dependent on the amount of averaging, and other factors). LODs depend on the averaging times, ionization mechanism, and instrument dependent background. LODs for n-alkanes studied here are calculated to be between 10 ppt for *n*-dodecane and 460 ppt for *n*-hexane. These LODs seem appropriate for studies such as pollution plumes or oil evaporation.

### 3.3.1 Dependence of CT sensitivity on ionization energy

The sensitivities to reaction by charge transfer for the *n*-alkanes was observed to be proportional to the chain length from n-pentane to *n*-dodecane. This is due to the higher polarizability and stability of larger *n*-alkanes (Cao and Yuan, 2002), as shown by the exponentially negative association between their normalized sensitivities and ionization energies (IE) (Fig. 6) (NSens= $4E+36x^{-37.33}$, $r^2$=0.97, n=9). The sensitivities to reaction by HA followed the same pattern as by CT (Fig. 5), also increasing with the chain length from *n*-heptane. PT reactions had the lowest sensitivities for *n*-alkanes in our study (Fig. 5). In fact, negligible or no PT products were found in this study for *n*-alkanes smaller than *n*-decane. Meanwhile a decrease in sensitivity to reaction by CT and HA was observed for *n*-tridecane, an increase in the PT sensitivity was observed when the carbon number increased. This trend was previously shown by SIFT studies for larger *n*-alkanes by the PT mechanism, where the reactions with $H_3O^+$ becomes more exothermic (Arnold et al. 1998).

### 3.3.2 Double hydride abstraction

Figure 7 demonstrates the observation of double hydride abstraction (DHA) products from *n*-alkanes. DHA/CT products are shown in Fig. 7, were DHA/CT percentages were more abundant for *n*-pentane and *n*-hexane. Little information has been reported on DHA mechanisms. Francis et al. (2007) reported DHA yields from ethane (15 %) and propane (5 %) in the presence of $O_2^+$, while dissociation was reported for higher *n*-alkanes (N*c*=4-9). Arnold et al. (1997) reported DHA in the reaction of *n*-octane (1 %) and isooctane (7 %) with $O_2^+$. Positive correlations of DHA for *n*-hexane and *n*-decane vs. $O_2^+$ (Fig. S5) were obtained, suggesting DHA is driven by $O_2^+$.

### 4 Conclusions

We explored and optimized mixed ionization mode conditions in the PTR-MS to enhance the detection of *n*-alkanes. Ionization using a combination of $O_2^+$ and $NO^+$, along with $H_3O^+$, allowed access to multiple ionization mechanisms with different sensitivities and specificities for *n*-alkanes. Low water flow (2 sccm), low E/N ratio (83 Td) and high $O_2^+$ ($U_{so}$=180V) provided the best overall conditions for analysis of *n*-alkanes. The charge transfer mechanism produced the maximum sensitivities for *n*-alkanes higher than *n*-hexane, followed by the hydride abstraction mechanism. Hydride abstraction was the most sensitive mechanism for *n*-hexane and *n*-pentane, followed by charge transfer and double hydride extraction. Sensitivities to charge transfer and hydride abstraction increased with carbon



chain length from *n*-pentane to *n*-dodecane. Proton transfer was not observed for *n*-alkanes with chain lengths from five to nine. From *n*-decane, the sensitivity to proton transfer increased with carbon chain length. Ionization by $O_2^+$

was highly sensitive through both charge transfer and double hydride abstraction, while with $NO^+$ only hydride abstraction was observed. No direct proton transfer was observed for *n*-alkanes but relatively small water cluster $[M.H_3O]^+$ formation was observed with a tendency to dissociate to $[M+H]^+$ species. We observed an exponentially negative association between the normalized sensitivities under charge transfer and molecular ionization energies due to polarizability and stability of the larger *n*-alkanes. The identification of a double hydride abstraction

mechanism by $O_2^+$ ionization is a new insight providing a novel selective and sensitive detection approach for measuring *n*-pentane and *n*-hexane with PTR-MS. The conditions determined in this study allow for the use of PTR-MS to sensitively measure *n*-alkanes in mixed ionization mode. This approach could also be extended to other alkane compounds like branched and cycloalkanes that constituent important mass fraction of petroleum derived products which are not measured with traditional PTR-MS.

*Acknowledgments*

AMO acknowledges financial support of UCMEXUS-CONACyT Program for the Postdoctoral Fellowship. This research was made possible by a grant from The Gulf of Mexico Research Initiative (GOMRI), as part of the Gulf of Mexico Integrated Spill Response Consortium (GISR) under contract SA12-09/GoMRI-006.

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



# Tables

Table 1. *N*-alkanes used in experiments. Formula, molecular weight (MW), monitored ions indicating hydride abstraction (HA), charge transfer (CT) and proton transfer (PT), isotopic contribution ($^{13}C/^{12}C$), Ionization energies (IE) and Proton affinities (PA). Water proton affinity: 166.5±2 kcal mol$^{-1}$ (Lias et al. 1984)

| # | Compound | Formula | MW, Da | HA, m/z | CT, m/z | PT, m/z | $^{13}C/^{12}C$ % | IE, eV | PA[c] Kcal/mol |
|---|----------|---------|--------|---------|---------|---------|------------------|--------|----------------|
| | | | | **Mechanism and Ions monitored** | | | | | |
| 1 | *n*-pentane | $C_5H_{12}$ | 72 | 71 | 72 | 73 | 5.6 | 10.28[a] | - |
| 2 | *n*-hexane | $C_6H_{14}$ | 86 | 85 | 86 | 87 | 6.7 | 10.13[a] | 160.7 |
| 3 | *n*-heptane | $C_7H_{16}$ | 100 | 99 | 100 | 101 | 7.8 | 9.93[a] | - |
| 4 | *n*-octane | $C_8H_{18}$ | 114 | 113 | 114 | 115 | 8.9 | 9.80[a] | 163.4 |
| 5 | *n*-nonane | $C_9H_{20}$ | 128 | 127 | 128 | 129 | 10.0 | 9.71[a] | - |
| 6 | *n*-decane | $C_{10}H_{22}$ | 142 | 141 | 142 | 143 | 11.1 | 9.65[a] | 165.0 |
| 7 | *n*-undecane | $C_{11}H_{24}$ | 156 | 155 | 156 | 157 | 12.2 | 9.56[a] | - |
| 8 | *n*-dodecane | $C_{12}H_{26}$ | 170 | 169 | 170 | 171 | 13.3 | 9.48[b] | 165.6 |
| 9 | *n*-tridecane | $C_{13}H_{28}$ | 184 | 183 | 184 | 185 | 14.5 | 9.42[b] | - |

[a] NIST chemistry web book (webbook.nist.gov), [b] Zhou et al. (2010), theoretical values calculated by density
functional theory at B3P86/6-31++G(d,p) level, [c] Hunter and East (2002).

Table 2. Sensitivity ratios for different mechanisms: [M$^+$] charge transfer, [M-H]$^+$ hydride abstraction and [M+H]$^+$ proton transfer. Dominant mechanism is shown in parenthesis.

| *n*-alkane | Optimized conditions 2 sccm, 83 Td, 180 V | Usual conditions 6 sccm, 122 Td, 60 V | Optimized conditions 2 sccm, 83 Td, 180 V | Usual conditions 6 sccm, 122 Td, 60 V |
|-----------|-------------------------------------------|---------------------------------------|-------------------------------------------|---------------------------------------|
| (HA) | [M-H]$^+$/[M+H]$^+$ | | [M-H]$^+$/M$^+$ | |
| *n-h*exane | 344.0 | [M+H]$^+$= n.d. | 1.9 | M$^+$ = n.d. |
| | | | | |
| (CT) | M$^+$/[M+H]$^+$ | | M$^+$/[M-H]$^+$ | |
| *n*-nonane | 50.5 | M$^+$, [M+H]$^+$ = n.d. | 2.4 | M$^+$, [M-H]$^+$ = n.d. |
| *n*-decane | 17.6 | [M+H]$^+$ = n.d. | 1.6 | 1.1 |
| *n*-undecane | 8.0 | [M+H]$^+$ = n.d. | 1.6 | 0.5 |
| *n*-dodecane | 3.5 | [M+H]$^+$ = n.d. | 1.6 | 1.0 |

n.d. – no detected




# Figures


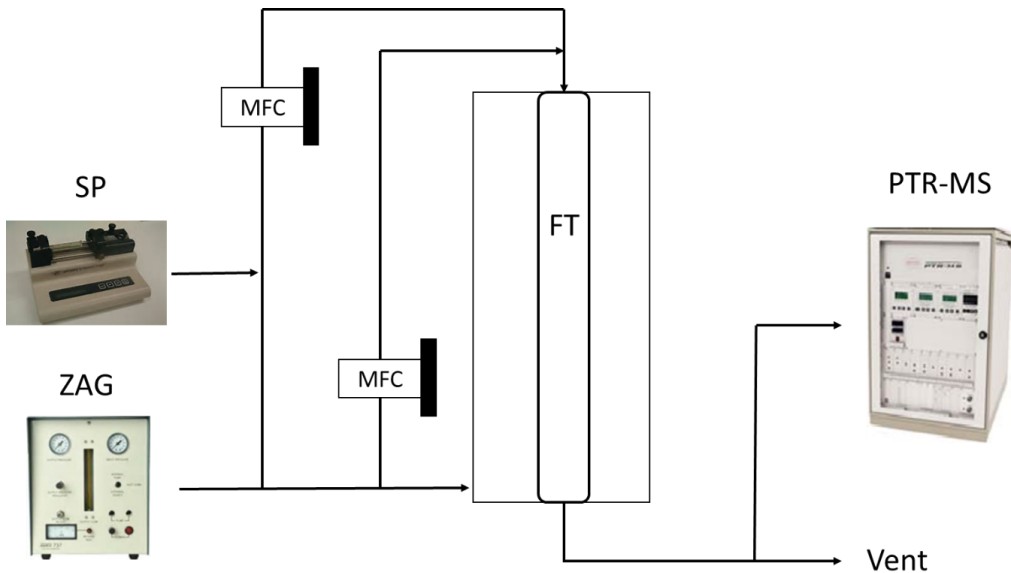

Figure 1. Schematic of the flow tube (FT) used to mix and dilute the *n*-alkanes coming from the syringe
pump (SP) with zero air (ZA), followed by analysis using proton transfer reaction mass
spectrometry (PTR-MS).







a

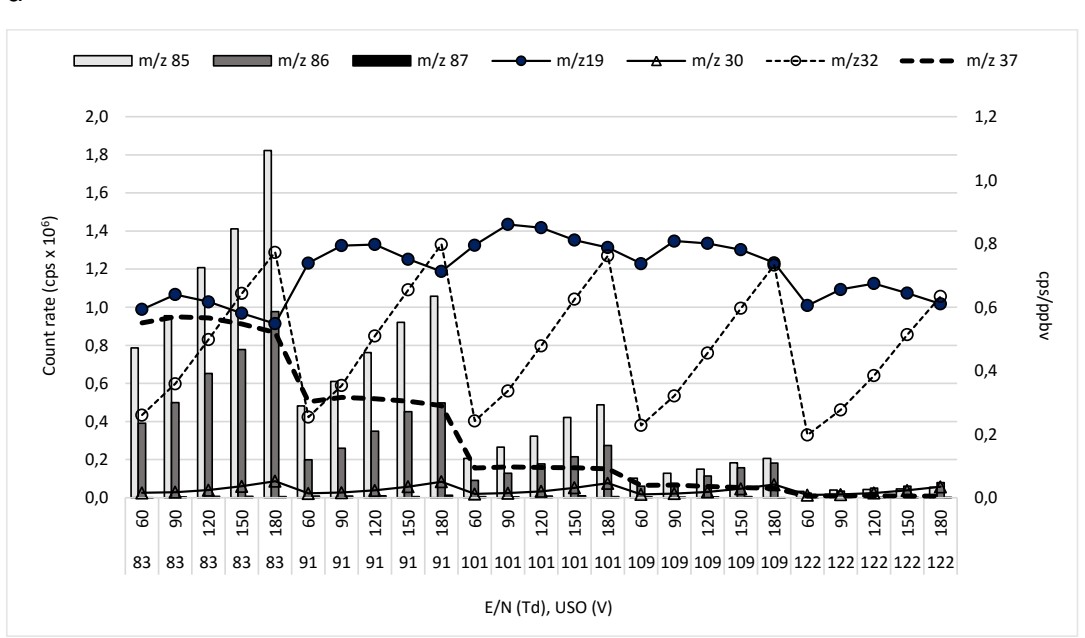

b

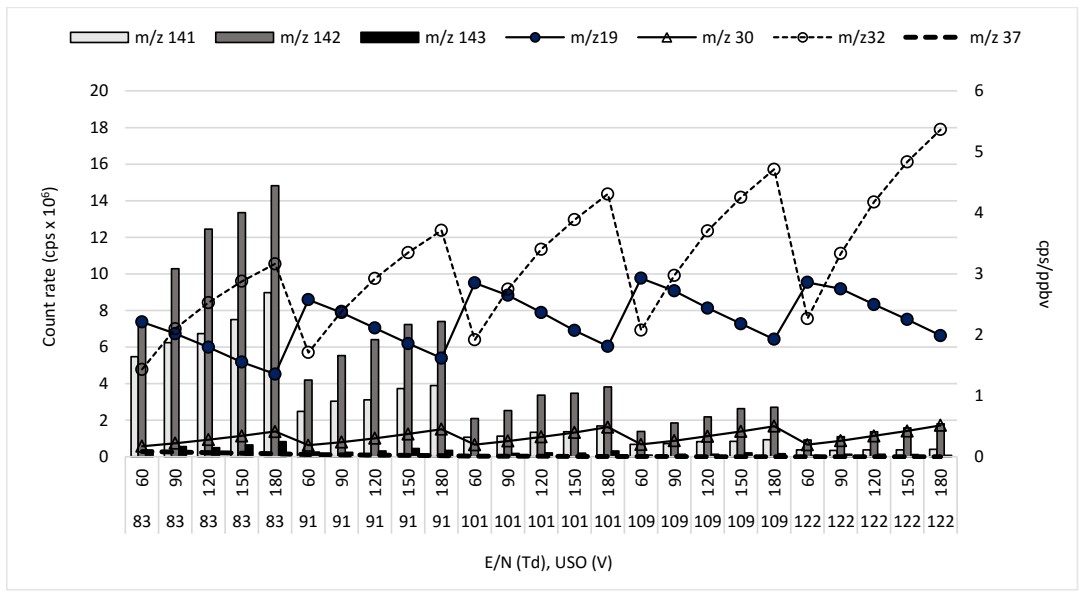

Figure 2. Intensity of signals (counts per second, cps) for $H_3O^+$ (m/z 19), $NO^+$ (m/z 30), $O_2^+$ (m/z 32), $H_2O.(H_3O)^+$ (m/z 37) and sensitivities (cps/ppbv) at low water flow (2 sccm) for **a.** *n*-hexane and **b.** *n*-decane. Data are shown for tests at five E/N ratios (83, 91, 101, 109 and 122 Td) and five $U_{SO}$ voltages (60, 90, 120, 150 and 180).






a

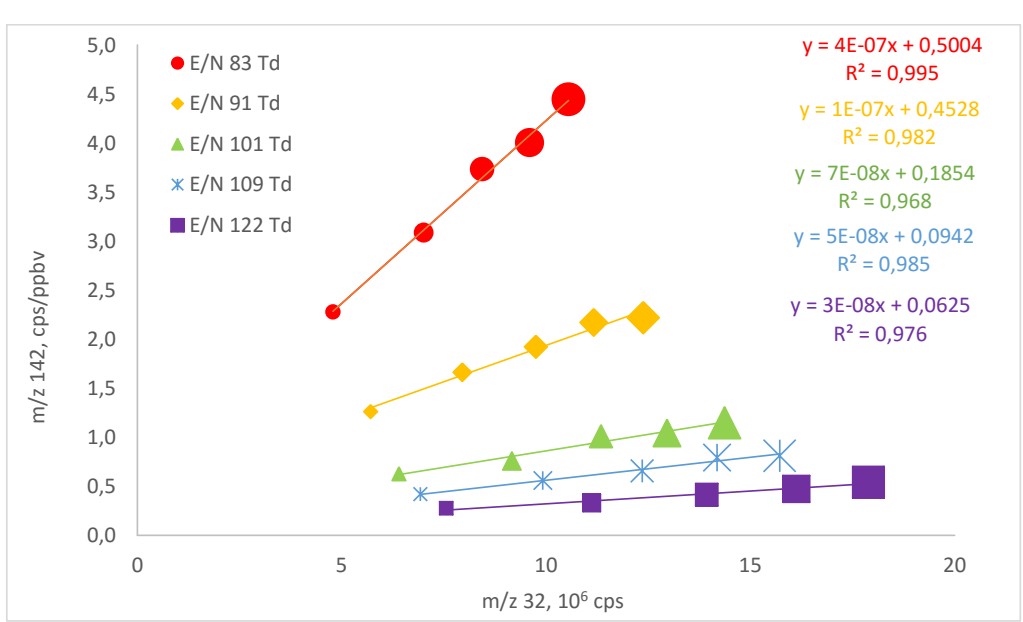

b

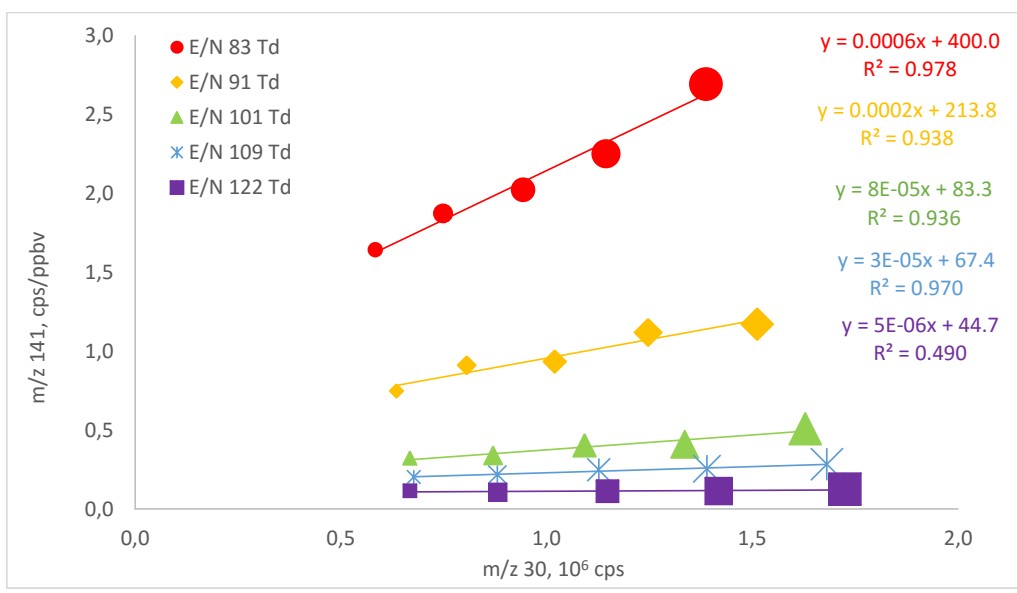

Figure 3. Intensities of (**a**) m/z 142 (M$^+$, charge transfer) vs. m/z 32 (O$_2^+$) for *n*-decane, and (**b**) m/z 141
([M-H]$^+$, hydride abstraction), vs. m/z 30 (NO$^+$) for *n*-decane. Marker size illustrates the U$_{SO}$
values: Smallest 60 V and largest 180 V.





a

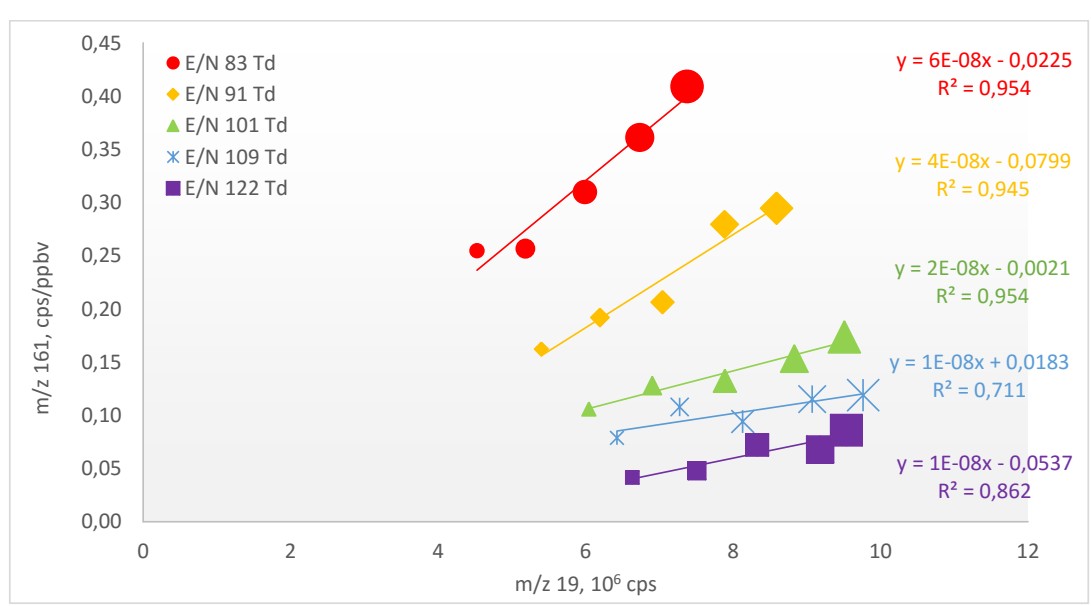

b

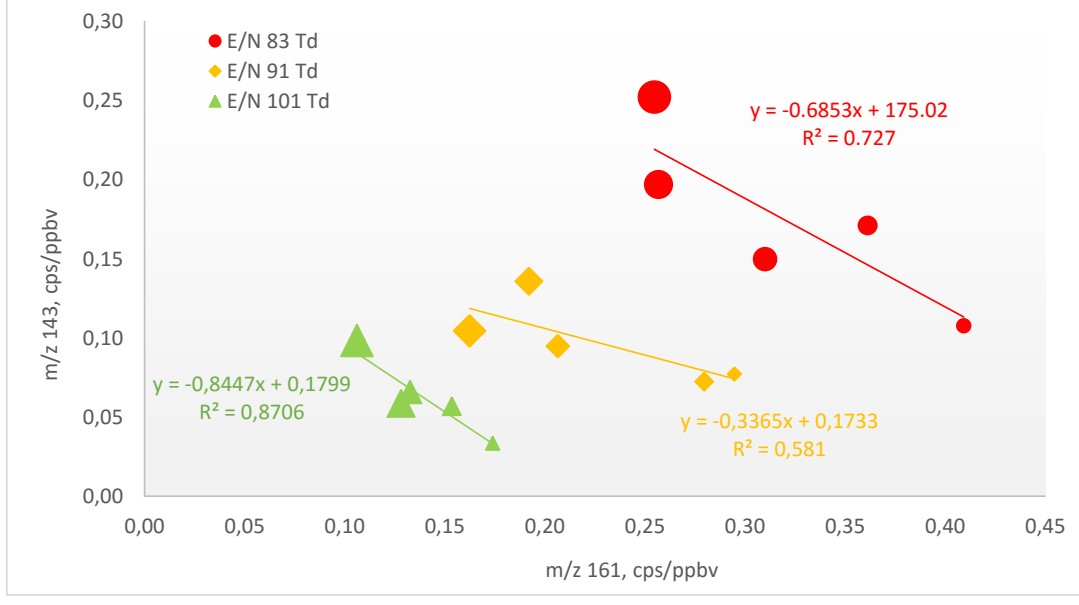

Figure 4. Intensities of the *n*-decane-water cluster (m/z 161, M.H$_3$O$^+$) vs. (a) hydronium (m/z 19, H$_3$O$^+$) and (b) proton transfer product (m/z 143, M+H$^+$). Marker size illustrates the U$_{SO}$: Smallest 60 V and largest 180 V. No correlations were observed at 109 and 122 Td.





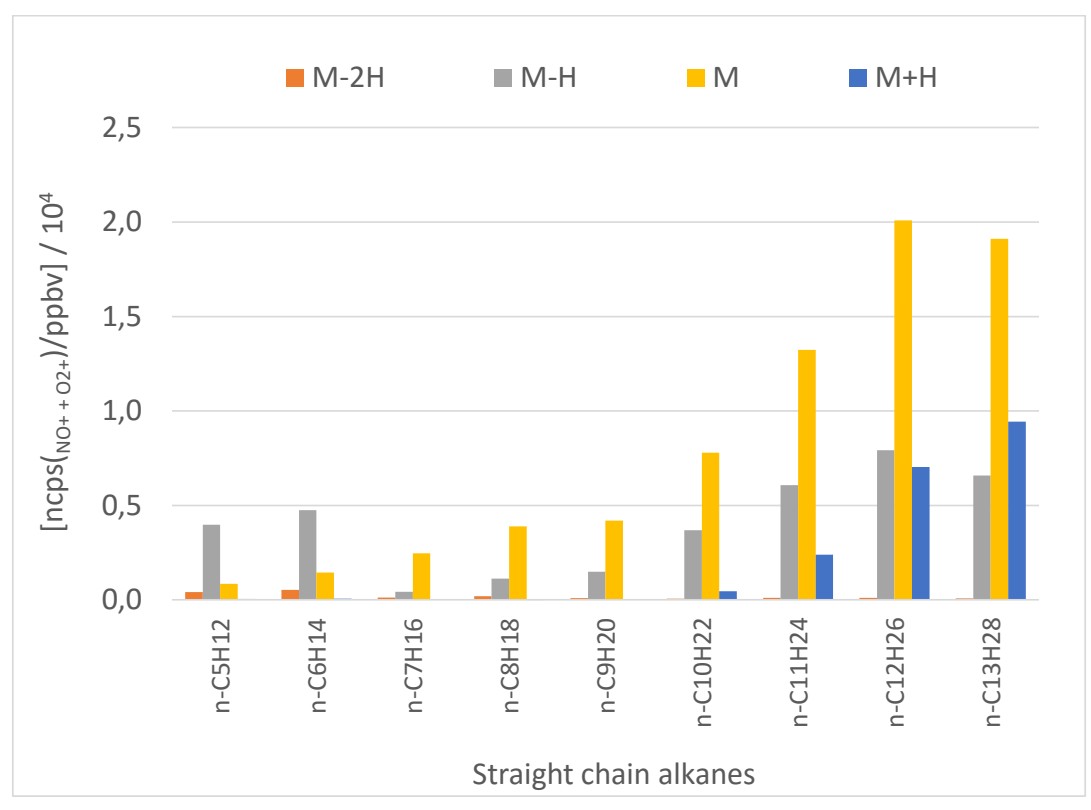


Figure 5. Sensitivities normalized to the sum ($NO^+ + O_2^+$) for straight chain alkanes detected by different mechanisms: double hydride abstraction (M-2H), hydride abstraction (M-H), charge transfer (M) and proton transfer (M+H).






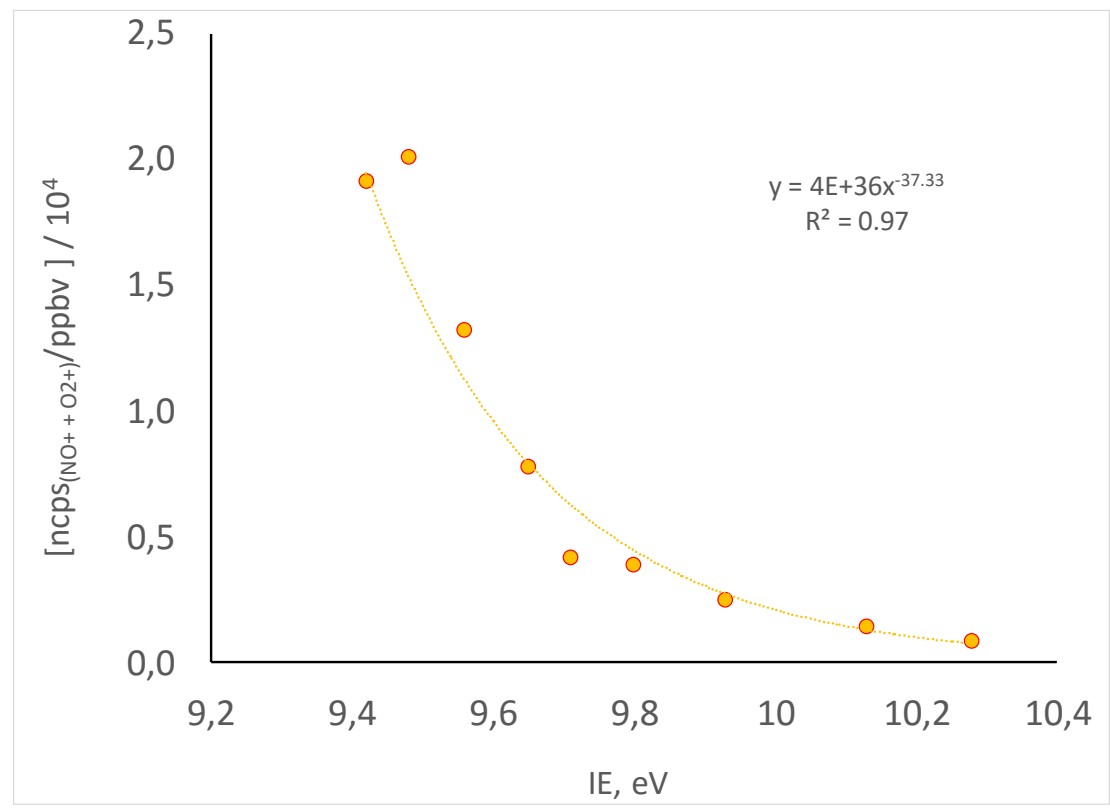

Figure 6.  Ionization energy vs. sensitivities normalized to the sum ($NO^+ + O_2^+$) (ncps/ppbv) for *n*-alkanes
        reacting by charge transfer.





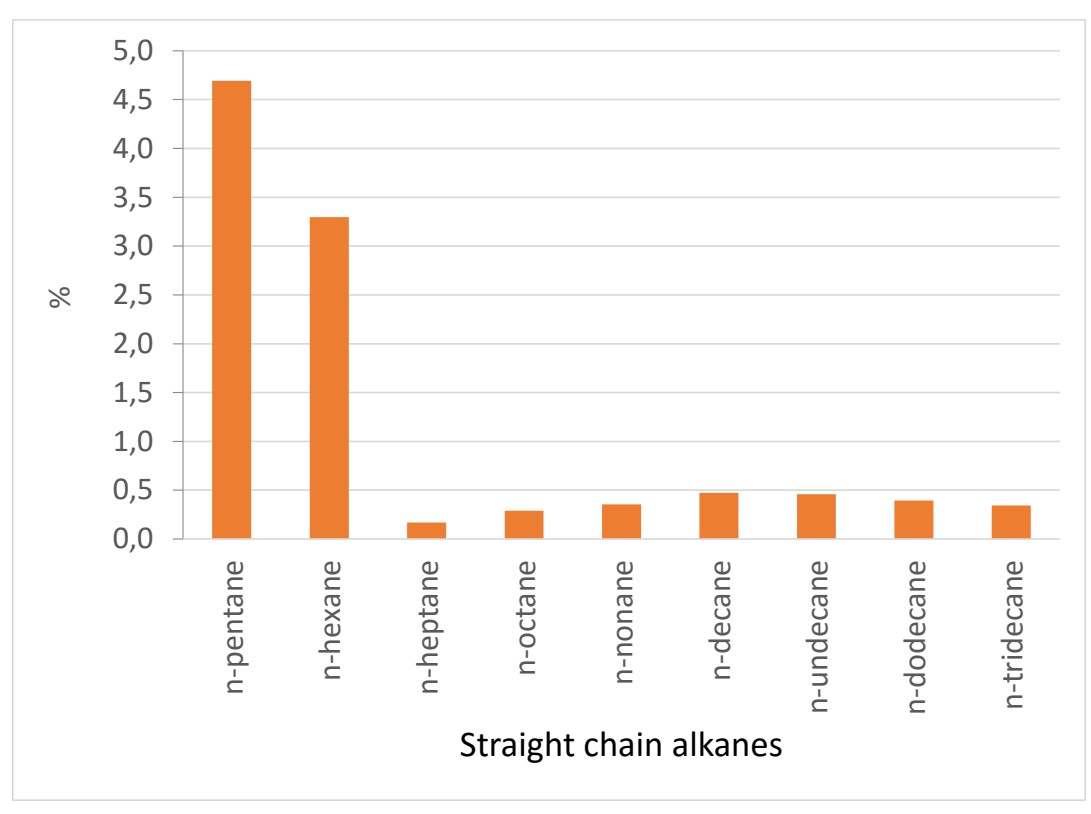

Figure 7. Contribution of double hydride abstraction (DHA) to charge transfer (CT). Calculated using sensitivities (ncps/ppbv) normalized to the sum $O_2^+ + NO^+$.

