# Peer review of "The next graphs in figure S1 show the variability at low water flow (2 sccm) of the key source ions comprising $H_3O^+$ (m/z 19), $O_2^+$ (m/z 32), $NO^+$ (m/z 30) and $H_2O.(H_3O)^+$ (m/z 37) as a function of E/N. Due to excessive water clusters at low E/N found at 6 sccm (HWF), the primary"

_Atmospheric Measurement Techniques, 2016_

## Referee Comment (RC1) · Anonymous Referee #1 · 11 Jun 2016

This paper describes a method for the detection of n-alkanes using a standard PTR-MS instrument with different settings. n-alkanes are not detectable with H3O+ primary ion chemistry, but n-alkane measurements are of increasing importance due to their emissions from oil&gas developments and crude oil evaporation and the authors mention crude oil evaporation as the main motivation for this work. A few recent papers have shown that NO+ primary ions are a promising method for detecting n-alkanes. In this paper a mixture of H3O+, O2+ and NO+ is used for primary ions, which results in various reaction channels and therefore multiple product ions for each n-alkane, severely complicating the detection. The paper also does not show measurement results of mixtures of n-alkanes or for example crude oil evaporation, so that possible overlap

from fragments of larger alkanes or other compounds cannot be judged from the data presented. I suspect that fragmentation and overlap will make this method not very useful. In addition, it is well known that O2+ reacts with alkanes, which is the main reaction channel used here, so I am wondering what is new here. If the manuscript can be improved to demonstrate the usefulness of this method, this paper might be acceptable, but certainly not in its current form. I have several major comments and some minor ones that need to be addressed. The major comments basically all deal with the usefulness and the novelty of the method as presented here.

Major comments:

- As I mentioned above, I am not convinced that this method as described here, is actually useful, not even for oil evaporation, where the contributions of other species besides alkanes is minimal. The main primary ion at the tested instrument settings for the reaction with the n-alkanes is O2+. H3O+ is also present, but does not react with the alkanes. Various groups, foremost D. Smith and co-workers, have investigated the use of O2+ over the years, including for n-alkanes, and O2+ has basically been abandoned due to strong fragmentation in the charge transfer reactions with most compounds, except maybe for NH3 (Norman et al 2007). Fragmentation from O2+ with n-alkanes has also been documented in several more recent papers; a good example is by Francis et al 2007. The main issue is that larger alkanes fragment onto the same masses, where smaller alkanes are detected, such that in more complex alkane mixtures none of the individual species can be detected unambiguously. Recently the use of NO+ has become more popular in the PTR-MS community and it was shown that NO+ is a very promising primary ion for alkanes, but still suffers from fragmentation resulting in compound overlap; especially the work by Inomata et al 2014 and Yamada et al 2015 but also a very recent paper by Koss et al AMTD 2016 discuss this issue. In the method used here, one not only has to deal with fragmentation, but also competing reaction channels from NO+ and H3O+. Fragmentation, the main complication in all the other papers, is basically ignored in this manuscript; only the unfragmented

products from proton transfer (PT), charge transfer (CT) and hydrogen abstraction (HA) are discussed. Fragments are mentioned briefly in the text and a mass spectrum of n-decane is shown in the SI, and those clearly show the strong fragmentation with this current setup. In my opinion, the fragmentation will be the most restricting factor to the usefulness of this setup and I think a rigorous discussion of the fragmentation patterns needs to be added to the manuscript. This should include the actual fragmentation patterns of all investigated alkanes, a discussion of the contribution of larger alkanes to smaller alkanes and a detailed comparison to the ample literature of the reaction products from NO+ and O2+.

- The authors describe that their PTR-MS was not equipped with SRI and therefore it was easier to de-tune the ion source to generate NO+ and O2+ ions. This seems like a weak argument to me. To make NO+ one has to only replace the water reservoir with a synthetic air tank on the PTR-MS. This takes less than an hour to do. The optimum instrument setup for NO+ is well described by Jordan et al 2009 and Inomata et al 2014. A better justification is needed for why this method of producing clean NO+ was not used.

- With the competition between PT, CT and HA each measured compound likely has different product ions. How can you know for an unknown compound mixture which is the dominant reaction channel? Just a simple example: mass 71 is the HA product from n-pentane, PT product from MVK+MACR, and PT product from cyclopentane, which is an important constituent of crude oil (Yuan et al 2014). In addition mass 71 is a fragment of larger alkanes. Were you planning to use an analysis similar to Figure 3 for identification?

- Nothing demonstrates the usefulness of a method better than an example. A good one would be to calibrate all the n-alkanes simultaneously and show that you can identify and quantify all the alkanes correctly with the method described here. A mass spectrum of a crude oil sample with all the n-alkane peaks identified would be also very useful. A comparison with a standard method such as GC-MS would be even better.

An example is really needed!

Minor comments:

Line 55: I think it would be appropriate to use references from other groups for large alkane sources and abundances as well.

Line 83: Also here it would be appropriate to add a few more references such as Yamada et 2015, Knighton et al 2009, Koss et al 2016.

Line 92: There are several newer PTR-MS references that should be added here as well.

Line 117: You are varying all primary ions not only $O_2^+$.

Line 173-180: The description of Figure 2 does not include a discussion of the primary ions that are shown in the Figure. There also seems to be a mistake in the y-axis in Fig 2a. I also assume that Fig 2a shows the HWF case and Fig 2b the LWF case, at least for the primary ions. This needs to be explained.

Line 179: Here would be a very good place to start discussing the fragmentation products.

Line 188: How are the isotopes taken into account? Did you subtract the isoptopic signal from the PT channel for this ratio? If not, this should be done.

Line 196: A similar discussion should be added for PT. It is mentioned in several places in the text that the PA of alkanes is too low to react with $H_3O^+$, but it should be discussed more explicitly.

Line 204: Have you looked at cycloalkanes as well? They are a large fraction of crude oil and undergo slow, but measureable PT. Those will have a large influence on your measurements. Do you know how they will vary with E/N?

Line 265: I would call R7 a ligand switching reaction. This is a prominent channel at

low E/N as you have here.

Line 292: Figures 3 and 5 are somewhat inconsistent. The sensitivity shown in Figure 5 is higher than in Figure 3. Is this because of different water flow used 1sccm vs 2sccm?

Line 297: What are the averaging times that you used for the detection limits?

Figure 5: The y-axis is divided by 10e4. Where does that come from? This is not consistent with the rest of the manuscript.

---

## Author Comment (AC1) · 16 Jun 2016

We thank anonymous referee #1 for reading the manuscript and providing many useful comments that we will address in the revised version. We strongly believe that this work is valuable and can be useful for the CIMS community.

1. This paper describes a method for the detection of n-alkanes using a standard PTR-MS instrument with different settings. n-alkanes are not detectable with H3O+ primary ion chemistry, but n-alkane measurements are of increasing importance due to their emissions from oil&gas developments and crude oil evaporation and the authors mention crude oil evaporation as the main motivation for this work.

[Figure]

In this paper we focus on characterization of the ability to detect n-alkanes under different mechanisms available in the mixed ionization mode. We did not focus on crude oil evaporation, but one motivation for this work was developing an analytical method suitable for the detection of a broader suite of compounds than is possible with any of the common pure ionization modes possible with PTR-MS. This has never been done before and we found that the mixed mode can be sensitive not only to n-alkanes but also to most VOCs which can be protonated by H3O+. The contribution from the NO+ channel was relatively small (∼10%), but the ions sensitive to NO+ can also be detected if they are highly abundant. The main benefit of the mixed mode is that no switching is required making the real time data available simultaneously. While this mode is more challenging to interpret, we explicitly emphasize these challenges such as fragmentation and interferences. Furthermore, we point to where this method could be useful (e.g. matrices dominated by n-alkanes and aromatics such as fossil fuel emissions).

2. A few recent papers have shown that NO+ primary ions are a promising method for detecting n-alkanes. In this paper a mixture of H3O+, O2+ and NO+ is used for primary ions, which results in various reaction channels and therefore multiple product ions for each n-alkane, severely complicating the detection.

We agree that NO+ is a very promising mode, but as nicely reviewed by Koss et al., 2016 this mode is only useful for branched chain and long chain alkanes. We think that our development is highly complementary to pure O2+, NO+, and H3O+ modes. We demonstrated that O2+ is the best primary ion to detect n-straight alkanes > nC5H12 via charge transfer as opposed to hydride abstraction. The detection of n-straight chain alkanes is less sensitive via hydride abstraction than by charge transfer (a factor of 2-3), as illustrated in figure 5. The detection is generally complex but this is also true to some extent for pure modes where the impurity ions can be 1-2 orders of magnitude lower, while the measured concentrations often span 6 or more orders of magnitude. The pure mode in PTR-MS should still be regarded as mixed mode because there remain

contributions and interferences that will necessarily depend on the studied matrix and concentrations. For example, if there is 1% of O2+ in the pure NO+ mode, 100 ppb of pentane in a plume can make a relatively large signal due to O2+, which would interfere with cyclopentane. We would also like to point out that while individual n-alkanes are most uniquely detected by GC-MS, the mixed mode offers a good time-dependent tracer for a sum of all alkanes which can serve as a link between detailed chemical composition of alkanes (GC-MS) and high time resolution of the varying emissions or concentrations (PTR-MS in mixed mode).

3. The paper also does not show measurement results of mixtures of n-alkanes or for example crude oil evaporation, so that possible overlap from fragments of larger alkanes or other compounds cannot be judged from the data presented. I suspect that fragmentation and overlap will make this method not very useful.

While crude oil evaporation was not the focus, we will add examples of measuring mixtures of alkanes and estimating the concentrations of n-alkanes. In our method, we take into account n-alkane fragmentation by using the calibrated fractions from individual n-alkane experiments. We show that the larger alkane fragments are successfully subtracted from the lower alkanes ions, and demonstrating that this method works reasonably.

4. In addition, it is well known that O2+ reacts with alkanes, which is the main reaction channel used here, so I am wondering what is new here.

It is well known that O2+ reacts with alkanes, but except for SIFT studies (e.g. Francis et al., 2007), the O2+ mode and its mechanisms have not been evaluated on PTR-MS. We find that this mode is indeed very useful for alkane detection and we show optimizations to achieve high sensitivity, and suppress excessive fragmentation. We also explore in great detail which mechanisms produce the highest sensitivities as a function of carbon number. For n-hexane and lower n-alkanes the most sensitive mechanism was hydride abstraction, while for higher n-alkanes, the most sensitive mechanism was

charge transfer. This is new information that will be valuable to the broader community. While our in-depth characterization of O2+ ionization mechanisms is presented with the goal of O2+ detection, it should be clear that this detailed characterization will also be useful to those who will regard O2+ as an impurity in other detection schemes. Our characterization of the O2+ ionization will thus enhance interpretation of data for complex mixtures using any ionization scheme. In addition, the mixed mode utilizes H3O+, and to a smaller extent NO+ for enabling measurement of a much broader range of compounds simultaneously.

5. If the manuscript can be improved to demonstrate the usefulness of this method, this paper might be acceptable, but certainly not in its current form. I have several major comments and some minor ones that need to be addressed. The major comments basically all deal with the usefulness and the novelty of the method as presented here. Major comments: - As I mentioned above, I am not convinced that this method as described here, is actually useful, not even for oil evaporation, where the contributions of other species besides alkanes is minimal.

We strongly disagree with this comment. In addition to alkanes, oil contains a large fraction of aromatics, so the mixed mode is much more useful than the pure modes because it allows for simultaneous tracking of alkanes and aromatic compounds without the need for ion switching. Again, we did not focus on determining alkanes in oil. Our main focus was on determining the mechanisms for detection of alkanes and optimizing the use of mixed-mode conditions. We are confident that the mixed mode, first proposed here, is novel and can be very useful for numerous applications where air is dominated by VOCs such as aromatics, and alkanes. We include the example of a complex alkane+VOC matrix which shows that the mixed mode can be promising also for complex matrices although not all (e.g. biogenics). There are very few studies on O2+ detection of alkanes and the mixed mode has never been proposed before. The double hydride abstraction mechanism in O2+ is a particularly novel finding in our study and more specific for the detection of some alkanes. We have not only described

optimizations and characterizations of this method, but the revised paper further emphasizes the novelty and usefulness of the method as well as highlights the known and less known challenges more clearly.

6. The main primary ion at the tested instrument settings for the reaction with the n-alkanes is O2+. H3O+ is also present, but does not react with the alkanes. Various groups, foremost D. Smith and co-workers, have investigated the use of O2+ over the years, including for n-alkanes, and O2+ has basically been abandoned due to strong fragmentation in the charge transfer reactions with most compounds, except maybe for NH3 (Norman et al 2007). Fragmentation from O2+ with n-alkanes has also been documented in several more recent papers; a good example is by Francis et al 2007.

We directly addressed the expected challenges of O2+ ionization that have limited previous investigations by systematically studying a very large and comprehensive matrix of instrumental (voltages etc.) and experimental (humidity etc.) conditions. It is true that the H3O+ does not react with alkanes, which is desirable in this study. The role of H3O+ in the mixed mode is not the ionization of alkanes but of the other compounds. Crude oil is just an example, but there can be other environments where alkanes and other classes of VOCs (e.g. aromatics) need to be analyzed simultaneously. The number of studies using O2+ ionization is actually extremely small relative to those on NO+ and H3O+ and almost exclusively comes from the SIFT community, where strong fragmentation is more difficult to suppress or account for. For example, Francis et al., 2007 used CIMS and focused only on short alkanes. We comprehensively evaluate the ionization mechanisms available in the mixed mode and increase understanding of theO2+ mechanism using a PTR-MS instrument. As mentioned earlier, we also characterized the mechanisms responsible for the highest sensitivities as a function of the carbon number and point out that hydride abstraction dominates the sensitivity for n-hexane and smaller hydrocarbons, while charge transfer is the most sensitive mechanism for larger alkanes.

7. The main issue is that larger alkanes fragment onto the same masses, where smaller

alkanes are detected, such that in more complex alkane mixtures none of the individual species can be detected unambiguously. Recently the use of NO+ has become more popular in the PTR-MS community and it was shown that NO+ is a very promising primary ion for alkanes, but still suffers from fragmentation resulting in compound overlap; especially the work by Inomata et al 2014 and Yamada et al 2015 but also a very recent paper by Koss et al AMTD 2016 discuss this issue. In the method used here, one not only has to deal with fragmentation, but also competing reaction channels from NO+ and H3O+.

The reviewer suggests that NO+ mode on its own is better than the mixed mode. This may be true, but only in some applications. NO+ is indeed promising for alkane detection, but has similar, although perhaps less pronounced, challenges in terms of fragmentation and multiple ionization mechanisms (CT, HA, NO+ association). We think that O2+ is highly complementary to NO+ because it is not sensitive to branched alkanes and is more sensitive to straight chain alkanes in the range where NO+ is not sensitive. Also, the alkanes follow a clear fragmentation pattern in O2+ mode, which we show can be accounted for.

8. Fragmentation, the main complication in all the other papers, is basically ignored in this manuscript; only the unfragmented products from proton transfer (PT), charge transfer (CT) and hydrogen abstraction (HA) are discussed. Fragments are mentioned briefly in the text and a mass spectrum of n-decane is shown in the SI, and those clearly show the strong fragmentation with this current setup. In my opinion, the fragmentation will be the most restricting factor to the usefulness of this setup and I think a rigorous discussion of the fragmentation patterns needs to be added to the manuscript. This should include the actual fragmentation patterns of all investigated alkanes, a discussion of the contribution of larger alkanes to smaller alkanes and a detailed comparison to the ample literature of the reaction products from NO+ and O2+.

We did not mean to ignore the fragmentation of alkanes. In contrast, we highlighted that this is an important issue which should be taken into account in complex alkane mixtures. While the earlier version of the manuscript focused on simple alkanes to characterize their detection, in response to this review, we include the algorithm to quantify the signal of individual alkanes in complex mixtures by subtracting the fragment fractions. We show that this method works reasonably well for a mixture of n-alkanes and is easy to apply.

9. The authors describe that their PTR-MS was not equipped with SRI and therefore it was easier to de-tune the ion source to generate NO+ and O2+ ions. This seems like a weak argument to me. To make NO+ one has to only replace the water reservoir with a synthetic air tank on the PTR-MS. This takes less than an hour to do. The optimum instrument setup for NO+ is well described by Jordan et al 2009 and Inomata et al 2014. A better justification is needed for why this method of producing clean NO+ was not used.

We are unsure which sentence was misleading. We certainly did not want to depreciate the value of SRI, which is very useful and available in recent versions of the instruments sold by Ionicon. In fact, we believe the mixed method can be made even more useful for instruments with SRI, where the proportions of each ionization agent can be adjusted to maximize detectability in complex matrices. In addition, we never intended to double the effort on the pure NO+ mode, which has nicely been reviewed recently by Koss et al. and other groups. Instead, we focused on the novel mixed ionization mode featuring O2+, H3O+ and NO+ simultaneously. We think that this development is unprecedented and highly innovative. Of course we fully recognize that this method is not free from the challenges that are present in other adaptations of PTR-MS instruments as well.

10. With the competition between PT, CT and HA each measured compound likely has different product ions. How can you know for an unknown compound mixture which is the dominant reaction channel? Just a simple example: mass 71 is the HA product from n-pentane, PT product from MVK+MACR, and PT product from cyclopentane, which is an important constituent of crude oil (Yuan et al 2014). In addition mass 71 is a fragment of larger alkanes. Were you planning to use an analysis similar to Figure 3

for identification?

We found typically no or little competition between PT, CT, and HA for most ions. PT is not applicable to alkanes less than C10 and the sensitivity of cyclopentane (and its relative abundance in common mixtures such as oil) is very low compared to other alkanes. In addition, MVK+MAC is very low far away from biogenic sources and is only an issue for PTR-MS systems using a quadrupole detector. A potential interference with cyclopentane and pentane needs to be taken into account also for pure NO+ where O2+ (even if reduced to a few %) can have an important interference from much more abundant alkanes. Overall, detection of most compounds is dominated by a single ionization mechanism. Thus, overlap is mainly a problem when detecting compounds with very different concentrations. This can be true for any ionization scheme with reagent-ion impurities. Our comprehensive study of the ionization schemes at a minimum allows us to determine the major components of a sample and may often allow much more detailed characterization than has previously been possible.

11. Nothing demonstrates the usefulness of a method better than an example. A good one would be to calibrate all the n-alkanes simultaneously and show that you can identify and quantify all the alkanes correctly with the method described here. A mass spectrum of a crude oil sample with all the n-alkane peaks identified would be also very useful. A comparison with a standard method such as GC-MS would be even better. An example is really needed!

We thank the reviewer for this useful suggestion. We agree that the example of more complex mixture, and comparison with GC-MS method, would be useful for the readers. In fact, we did this work already but were planning to include it to another manuscript. In response to this comment, we will add to the manuscript the quantification of n-alkanes in crude oil using GCxGC-MS and show a simple method how the fragment contributions can be accounted for.

Minor comments: Line 55: I think it would be appropriate to use references from other

groups for large alkane sources and abundances as well. We now include more references.

Line 83: Also here it would be appropriate to add a few more references such as Yamada et 2015, Knighton et al 2009, Koss et al 2016. These references are now included.

Line 92: There are several newer PTR-MS references that should be added here as well. These have been added.

Line 117: You are varying all primary ions not only O2+. This has been now made clear.

Line 173-180: The description of Figure 2 does not include a discussion of the primary ions that are shown in the Figure. There also seems to be a mistake in the y-axis in Fig 2a. I also assume that Fig 2a shows the HWF case and Fig 2b the LWF case, at least for the primary ions. This needs to be explained. The primary ions and water flow cases are now clearly explained.

Line 179: Here would be a very good place to start discussing the fragmentation products. As suggested, we start discussing the fragmentation products here.

Line 188: How are the isotopes taken into account? Did you subtract the isoptopic signal from the PT channel for this ratio? If not, this should be done. As we wrote in p. 4, lines 127-128 and 146-147, the 13C isotopic signals were subtracted from the preceding ions.

Line 196: A similar discussion should be added for PT. It is mentioned in several places in the text that the PA of alkanes is too low to react with H3O+, but it should be discussed more explicitly. We implemented this suggestion in the revised manuscript.

Line 204: Have you looked at cycloalkanes as well? They are a large fraction of crude oil and undergo slow, but measureable PT. Those will have a large influence on your measurements. Do you know how they will vary with E/N? Yes, we did measurements

on cycloalkanes. As the reviewer noted, PT ions have very low sensitivity and in some cases suffer interference by HA from alkanes. This is a challenge for both NO+, O2+ and the mixed mode. We focus on the most abundant compounds such as n-alkanes, and aromatics. Because, cycloalkanes and alkanes fragment differently, one way to discriminate between cycloalkanes and alkanes would be to vary the E/N ratio but we do not want to focus on this in this manuscript.

Line 265: I would call R7 a ligand switching reaction. This is a prominent channel at low E/N as you have here. The manuscript has been updated accordingly

Line 292: Figures 3 and 5 are somewhat inconsistent. The sensitivity shown in Figure 5 is higher than in Figure 3. Is this because of different water flow used 1sccm vs 2sccm? The water flow rate affected the absolute count rate of primary ions. The scales in these figures are not the same. Figure 3 shows the absolute sensitivities while Figure 5 shows the normalized sensitivities to sum (O2+, NO+). Normalized sensitivities resulted in more consistent values for the CT reactions.

Line 297: What are the averaging times that you used for the detection limits? The average time was 0.2 s

Figure 5: The y-axis is divided by 10e4. Where does that come from? This is not consistent with the rest of the manuscript. In p8 lines 281-290, we explained that normalized sensitivity (ncps/ppbv) is recommended to account for the variability of levels of O2+ due to changing water flow in PTR-MS instruments (Jobson et al., 2005; Warneke et al., 2001). We calculated the normalized sensitivities to sum (O2+, NO+) to account for the variability between experiments. Because the response in signal of NO+ and O2+ are extremely high compared to response of the n-alkanes, the scale in y-axis was divided by 104

Once again, we thank the reviewer for stimulating the discussion which resulted in further improvement of the manuscript.

References: Francis, G. J., Wilson, P.F., Milligan, D.B., Langford, V.S., McEwan, M.J..: GeoVOC: A SIFT-MS method for the analysis of small linear hydrocarbons of relevance to oil exploration. International Journal of Mas Spectrometry, 268(1):38-46, 2007. Knighton, W. B., Fortner, E. C., Herndon, S. C., Wood, E. C., and Miake-Lye, R. C.: Adaptation of a proton transfer reaction mass spectrometer instrument to employ NO+ as reagent ion for the detection of 1,3-butadiene in the ambient atmosphere, Rapid Commun. Mass Sp., 23, 3301-3308, 10.1002/rcm.4249, 2009. Koss, A. R., Warneke, C., Yuan, B., Coggon, M. M., Veres, P. R., and de Gouw, J. A.: Evaluation of NO+ reagent ion chemistry for on-line measurements of atmospheric volatile organic compounds, Atmos. Meas. Tech. Discuss., doi:10.5194/amt-2016-78, in review, 2016.

---

## Referee Comment (RC2) · Anonymous Referee #2 · 17 Jun 2016

The paper describes a mixed mode operation of a standard PTR-MS instrument (i.e., H3O+, NO+ and O2+ primary ions present at the same time) to detect n-Alkanes. I agree with reviewer #1 that it will be extremely difficult to unambiguously detect these compounds in the real atmosphere using the approach described in the paper. However, I find the nicely conducted experimental work very useful for the PTR-MS community: Since a PTR-MS instrument always operates in a mixed mode (primary ion purity hardly ever reaches 95 %), information of product ions from n-Alkanes from the different precursor ions is valuable to many users, especially when interpreting PTR-TOF data from field campaigns. I strongly encourage the authors to include a table or figures in the paper or supplementary information, showing the full lists of fragment ions

and their branching ratio for each compound tested. If possible, corrected for mass-dependent transmission efficiy of the mass spectrometer (if not, please state clearly). After including this information and real world examples as suggested by reviewer #1 and held out in prospect by the authors' response, I find the paper definitely acceptable in AMT.

However, the phrasing in some parts of the paper (listed among other comments below) seem to be too optimistic in terms of atmospheric detectability of these compounds.

Additional comments:

Line 44: Hansel et al. 1995 is not the right citation here, back then the LOD was in the low ppbv range; a newer Ref would be appropriate

Line 63: According to table 1, the PA of none of the n-Alkanes listed is higher than the PA of water. There is some inconsistency here. A short discussion about sensitivities for compounds having only slightly higher PA's than water (like Formaldehyde, e.g.) and their water vapour dependencies due to backward reactions would be appropriate here.

Line 92: I cannot imagine better references describing "operation and fundamental principles of PTR-MS" here and therefore disagree with Reviewer #1

Line 103: To produce NO+ primary ions, usually a mixture of O2 and N2 is used as described in Karl et al, not pure NO

Line 128: "monitored ions" is misleading; as I understand, you made full mass scans; so you monitored all ions, including fragments. Giving only M-1, M-2, M and M+1 ions in the table, without mentioning that these ions were not the most prominent ones in many cases, is a bit of a whitewash here.

Line 170: The use of the term saturated is misleading here

Line 279: "H3O+ = 1 sccm"

Line 297: The phrasing seems too optimistic here, since it implies that these alkanes can be detected at these concentrations in the atmosphere, which I seriously doubt because of interferences. After the authors include results from mixtures, the LoD should be given for compounds in the presence of other n-Alkanes, rather than obtained from zero-air (as examples, demonstrating the range of detectabilities).

Line 330: the term "highly sensitive" should be avoided here; in PTR-MS community this terms is usually used for compounds being ionized at a kinetic rate, which is not the case here Figure 5: . . . is really hard to interpret; I agree with reviewer #1 and I am not convinced by the answer from the authors to this comment. Ncps/ppbv should always be used the same way: cps/ppbv for 1e6 primary ions per second; primary ions are these ion species which lead to the specific product ions (NO+ + O2+ for M; H3O+ for M+H etc.)

---

## Author Comment (AC2) · 30 Jul 2016

We thank the anonymous referee #2 for reading the manuscript and providing many useful comments which we have now addressed in the revised version.

1. The paper describes a mixed mode operation of a standard PTR-MS instrument (i.e., H3O+, NO+ and O2+ primary ions present at the same time) to detect n-Alkanes. I agree with reviewer #1 that it will be extremely difficult to unambiguously detect these compounds in the real atmosphere using the approach described in the paper. However, I find the nicely conducted experimental work very useful for the PTR-MS community: Since a PTR-MS instrument always operates in a mixed mode (primary ion purity hardly ever reaches 95 %), information of product ions from n-Alkanes from the

different precursor ions is valuable to many users, especially when interpreting PTR-TOF data from field campaigns. I strongly encourage the authors to include a table or figures in the paper or supplementary information, showing the full lists of fragment ions and their branching ratio for each compound tested. If possible, corrected for mass dependent transmission efficiy of the mass spectrometer (if not, please state clearly). After including this information and real world examples as suggested by reviewer #1 and held out in prospect by the authors' response, I find the paper definitely acceptable in AMT.

We appreciate these useful comments and we are pleased that the reviewer is finding our work useful for the PTR-MS community and acceptable in AMT. As suggested, we will include the tables with the fragment ions and their branched ratios. We agree that a typical mode of PTR-MS is also a mixed mode due to other reagent ion impurities and this is what the community often does not realize. Unlike with GC, a fully unambiguous identification is a general issue with CIMS and in particular when compounds are fragmenting it can very be difficult to discern signals for individual molecules but we are now showing the examples of the mixtures of alkanes and we are bracketing the potential bias to estimate the concentrations of n-alkanes, where larger alkane fragments are successfully subtracted from the lower alkanes ions using a calibrated fragmentation algorithm.

2. However, the phrasing in some parts of the paper (listed among other comments below) seem to be too optimistic in terms of atmospheric detectability of these compounds. Additional comments: Line 44: Hansel et al. 1995 is not the right citation here, back then the LOD was in the low ppbv range; a newer Ref would be appropriate.

We generally agree that the atmospheric detectability using the mixed mode is challenging. However, relative to early instruments (Hansel et al. 1995) with lower ppb range detection limits for alkanes, the sensitivity to alkanes in the mixed mode is quite impressive (of the order of tens cps per ppb) and the detection limits in the lower ppt range. This sensitivity is only within a factor of 2-3 lower than the sensitivity of the pure

mode but must be at cost of more complicated spectra and therefore the detection limit will be disproportionally higher. The higher effective detection limits are due to interferences with fragments which have to be accounted for. Overall, we hope that our research is useful to keep in mind the alkane ions which may be present in the pure modes if alkane concentrations are relatively high and possibility of usefulness of the mixed mode in studies of evaporation or alkanes in pollution plumes. We rephrased the parts which may have been interpreted as overly optimistic. We also updated the references with Karl et al. 2012.

3. Line 63: According to table 1, the PA of none of the n-Alkanes listed is higher than the PA of water. There is some inconsistency here. A short discussion about sensitivities for compounds having only slightly higher PA's than water (like Formaldehyde, e.g.) and their water vapour dependencies due to backward reactions would be appropriate here. We rephrase the sentence: "However, proton transfer reaction products are mostly unobserved for alkanes because the proton affinity of most alkanes is lower than that of water or in case of longer alkanes ($C \geq 10$) it is only very slightly higher than water leading to potential backward reactions"

4. Line 92: I cannot imagine better references describing "operation and fundamental principles of PTR-MS" here and therefore disagree with Reviewer #1. We thank the reviewer for this positive comment. We therefore keep these references, and we also add the references suggested by Reviewer #2.

5. Line 103: To produce NO+ primary ions, usually a mixture of O2 and N2 is used as described in Karl et al, not pure NO. Thanks.

6. Line 128: "monitored ions" is misleading; as I understand, you made full mass scans; so you monitored all ions, including fragments. Giving only M-1, M-2, M and M+1 ions in the table, without mentioning that these ions were not the most prominent ones in many cases, is a bit of a whitewash here. We replaced "monitored ions" with "corresponding ions". The same change was reflected in table 1.

7. Line 170: The use of the term saturated is misleading here ”primary ion was saturated” has been replaced with “which led to decline in primary ions”

8. Line 279: “H3O+ = 1 sccm” Done

9. Line 297: The phrasing seems too optimistic here, since it implies that these alkanes can be detected at these concentrations in the atmosphere, which I seriously doubt because of interferences. After the authors include results from mixtures, the LoD should be given for compounds in the presence of other n-Alkanes, rather than obtained from zero-air (as examples, demonstrating the range of detectabilities).

Further to our response to comment #2, the detection limits presented in the manuscript are the lower bounds for the detection limits which are valid if the atmosphere is dominated by a single alkane but we are now making it clear in the text that they may be significantly higher in complex alkane mixtures and this will depend how complex the mixture is and what the dominating alkanes are. 10. Line 330: the term “highly sensitive” should be avoided here; in PTR-MS community this terms is usually used for compounds being ionized at a kinetic rate, which is not the case here We refrained from using “highly sensitive”. However, the sensitivities are typically only up to 2-3 factors lower than those at a kinetic rate, so are relatively sensitive.

11. Figure 5: ... is really hard to interpret; I agree with reviewer #1 and I am not convinced by the answer from the authors to this comment. Ncps/ppbv should always be used the same way: cps/ppbv for 1e6 primary ions per second; primary ions are these ion species which lead to the speciïňĄc product ions (NO+ + O2+ for M; H3O+ for M+H etc.) We thank the reviewer for spotting these inconsistencies in the previous version of the manuscript. To be consisted with the PTRMS community terms, we are now consistently using the word "ncps" for the PT mechanism normalized to 1 million cps of H3O+. The CT and DHA mechanisms in the case of alkanes are due to O2+ so we show “weighted” sensitivities wcps (to distinguish from PTR terms) where are normalized to 1 million cps of O2+. In the case of HA mechanisms which can undergo

either O2+, or NO+ ionization or both, we show "weighted" sensitivities wcps which are normalized to 1 million cps of the sum of NO+ and O2+. We think this is appropriate because in the mixed mode, the relative proportion of NO+ to O2+ is relatively constant (∼10%) and correlated so this simple standardization of sensitivities does not require the knowledge of whether a given alkane is ionized by NO+ or O2+. We hope that the revised figure can be a useful summary of how these mechanisms distribute over the carbon numbers. To further increase the clarity we divided the figure into four panels. The PT panel uses normalized sensitivities to 1M of H3O+ (ncps/ppb), the HA panel shows weighted sensitivities panel normalized to 1E6 cps of the sum of O2+ and NO+ (wcps[O2+,NO+]), the CT and DHA panels use weighted sensitivities normalized to 1E6 cps of O2+ (wcps[O2+]). Another reason why the weighted sensitivities by relevant primary ions make sense is that in different configurations the absolute number of ions and therefore absolute sensitivities can be different or variable. This is one standardized way which should account for primary ion levels which are also provided for transparency, so that the user can easily un-normalize/un-weight.

Once again, we thank the anonymous reviewer 2 for these valuable comments which will significantly improve the manuscript.

References: Hansel, A.; Jordan, A.; Holzinger, R.; Prazeller, P.; Vogel, W. and Lindinger, W.: Proton transfer reaction mass spectrometry: on-line trace gas analysis at the ppb level, Int. J. Mass Spectrom. Ion Proc., doi:149/150, 609-619, 1995. Karl, T.; Hansel, A.; Cappellin, L.; Kaser, L.; Herdlinger-Blatt, I. and Jud, W.: Selective measurements of isoprene and 2-methyl-3-buten-2-ol based on NO+ ionization mass spectrometry, Atmos. Chem. Phys., 12, 11877-11884, doi:10.5194/acp-12-11877-2012, 2012.

Please also note the supplement to this comment:
http://www.atmos-meas-tech-discuss.net/amt-2016-64/amt-2016-64-AC2-supplement.pdf